# Biomass-Derived Plant Extracts in Macromolecular Chitosan Matrices as a Green Coating for PLA Films

**DOI:** 10.3390/jfb13040228

**Published:** 2022-11-07

**Authors:** Lidija Fras Zemljič, Tjaša Kraševac Glaser, Olivija Plohl, Ivan Anžel, Vida Šimat, Martina Čagalj, Eva Mežnar, Valentina Malin, Meta Sterniša, Sonja Smole Možina

**Affiliations:** 1Laboratory for Characterization and Processing of Polymers, Faculty of Mechanical Engineering, University of Maribor, Smetanova 17, 2000 Maribor, Slovenia; 2Materials Transformation Laboratory, Faculty of Mechanical Engineering, University of Maribor, Smetanova 17, 2000 Maribor, Slovenia; 3University Department of Marine Studies, University of Split, Ruđera Boškovića 37, 21000 Split, Croatia; 4Department of Food Science and Technology, Biotechnical Faculty, University of Ljubljana, Jamnikarjeva 101, 1000 Ljubljana, Slovenia

**Keywords:** PLA, packaging, active, biomass, chitosan, extracts, antioxidant, antimicrobial

## Abstract

Due to the growing problem of food and packaging waste, environmental awareness, and customer requirements for food safety, there is a great need for the development of innovative and functional packaging. Among these developments, the concept of active packaging is at the forefront. The shortcoming in this area is that there is still a lack of multifunctional concepts, as well as green approaches. Therefore, this work focuses on the development of active chemical substances of natural origin applied as a coating on polylactic acid (PLA) films. Biopolymer chitosan and plant extracts rich in phenolic compounds (blackberry leaves—Rubus fruticosus, needles of prickly juniper—Juniperus oxycedrus) obtained from plant biomass from Southeastern Europe were selected in this work. In order to increase the effectiveness of individual substances and to introduce multifunctionality, they were combined in the form of different colloidal structural formulations. The plant extracts were embedded in chitosan biopolymer particles and dispersed in a macromolecular chitosan solution. In addition, a two-layer coating, the first of a macromolecular chitosan solution, and the second of a dispersion of the embedded extracts in chitosan particles, was applied to the PLA films as a novel approach. The success of the coatings was monitored by X-ray photoelectron spectroscopy (XPS) and attenuated total reflection Fourier transform infrared spectroscopy (ATR-FTIR), and the wettability was evaluated by contact angle measurements. Scanning electron microscopy SEM tracked the morphology and homogeneity of the coating. Antioxidation was studied by DPPH and ABTS spectrophotometric tests, and microbiological analysis of the films was performed according to the ISO 22196 Standard. Desorption of the coating from the PLA was monitored by reducing the elemental composition of the films themselves. The successful functionalization of PLA was demonstrated, while the XPS and ATR-FTIR analyses clearly showed the peaks of elemental composition of the extracts and chitosan on the PLA surface. Moreover, in all cases, the contact angle of the bilayer coatings decreased by more than 35–60% and contributed to the anti-fogging properties. The desorption experiments, due to decrease in the concentration of the specific typical element (nitrogen), indicated some migration of substances from the PLA’s surface. The newly developed films also exhibited antioxidant properties, with antioxidant ABTS efficiencies ranging from 83.5 to 100% and a quite high inhibition of Gram-positive *Staphylococcus aureus* bacteria, averaging over 95%. The current functionalization of PLA simultaneously confers antifogging, antioxidant, and antimicrobial properties and drives the development of a biodegradable and environmentally friendly composite material using green chemistry principles.

## 1. Introduction

In the field of the development and use of products made of various materials with a high content of bio-based components, European industry can realize a great opportunity for a technological breakthrough in the use/reuse of biomass. The use of biomass directly contributes to the reduction of carbon dioxide in the atmosphere and thus, makes an important contribution to mitigating climate change. Compared to products derived from fossil sources, the remarkable advantage of bioproducts lies in their reproducibility, biodegradability, and/or compostability, and the possibility of their cascading use [1,2,3], which is one of the priorities for packaging materials industry.

In general, plastic packaging exhibits characteristics such as low production cost, good mechanical properties, and light weight, while providing quite a satisfactory oxygen barrier [4]. However, most of these polymers are of a petrochemical origin, and the increase in their consumption inevitably leads to socioeconomic problems, such as shortages and rising oil prices, and environmental problems, such as the generation and accumulation of solid waste that can take hundreds of years to decompose [5], thus being a considerable threat to the environment and human health. Although packaging has an important function in the global economy, producer responsibility policies remain scarce at the national and international levels. The results show that the transition to sustainable packaging in the food and beverage sector has been slow and uneven. Most company sustainability reports do not address plastic pollution, and those companies focus mainly on collection and recycling, rather than on sustainable packaging solutions aimed at systemic change [6,7]. Thus, new packaging solutions aim to reduce the environmental footprint (both in terms of biodegradability and the origin of raw materials for packaging). Additionally, there is a recognized need for so-called eco-innovative functional food packaging that directly benefits consumers by improving food shelf life, monitoring food quality, and ensuring food safety, while exerting a smaller environmental footprint than conventional food packaging. Therefore, these packaging materials must not only be biodegradable, but also have active functionality. Active systems can be successfully used to extend the shelf life of processed foods, and can be divided into adsorbent and releasing systems (e.g., oxygen scavengers, ethylene scavengers, liquid and moisture absorbers, taste and odor absorbers or releasers, antimicrobials, etc.) [8]. A useful strategy is to incorporate active ingredients into the matrix of the used packaging materials, or to apply coatings with appropriate functionality by surface modification. The latter option offers the advantage that the properties of the packaging material remain virtually unchanged. Active coating films with antioxidant and antimicrobial properties have gained more attention due to their ability to act at the interface, or transport bioactive compounds and release them to food in a controlled manner. They minimize oxidation reactions, preserve the microbiological and sensory aspects of foods and thus, extend the shelf life of foods, resulting in less food waste and the possibility for greater transportation distances [9]. In line with the vision of bio-based industries (BBI) and the concept of bioeconomy development, bio-based active components, especially those isolated from biomasses that introduce specific bioactive functionality while maintaining the biodegradability of the materials, represent a trend in the field of sustainable functional packaging materials. In this area, various biodegradable and natural substances are studied, including polyphenols (ginger essential oil, carvacrol, thymol), potent antioxidants (grape seed extract, yerba mate, rosemary, mint), and natural pigments (anthocyanins, curcumin, betalain, carotenoids) [10]. Polysaccharides, (alginate, carrageenan, chitosan, pectin, pullulan, starch, xanthan, etc.), and proteins (collagen, gelatin, whey and soy proteins, zeins, etc.), are attractive natural sources of antimicrobial and /or barrier polymers [10,11]. Due to its functional properties, chitosan is a promising bioactive polymer for food packaging applications [11]. Several studies on the combination of chitosan and plant extract as chitosan mixtures, or as coatings made of chitosan plant extracts, have been examined. Exemplary are chitosan with rosemary extracts, chitosan with galangal extract, and also chitosan in combination with essential cinnamon oil. In addition, studies have been conducted on combining chitosan with essential oil of clove, lavender, bergamot, citrus, tea tree, basil, oregano and thyme, and propolis. Most combinations show increased antimicrobial activity compared to pure chitosan, extending the shelf life of certain foods [12,13]. Although there are many combinations of chitosan with extracts, finding new combinations is still a challenge.

Moreover, lately the development of active components for biodegradable packaging has been strongly focused on the use and processing of agro-industrial plant products and by-products, with the aim of converting waste or by-products into value-added products, and thus reducing the use of conventional, non-renewable packaging and the integration of synthetic compounds. In this research area, there are still many opportunities to be exploited, especially since the European Union bioeconomy drivers and trends for 2030 and 2050 highlight that it will transform our use of biomass, leading to a decrease in the importance of bioenergy and biobased energy sources, and a greater focus on material recovery to be placed in the market for biobased materials, including innovative and advanced materials in the packaging sector [14].

We have previously studied primary chitosan (a biopolymer of natural origin derived from marine crustacean waste) as a potential coating, and found that it contained deficiencies due to pH-dependent protonation, and its antimicrobial activity was further limited in neutral and basic pH environments [15]. Therefore, in this study, we also included a quaternary chitosan whose amine groups were charged throughout the pH range, and therefore its activity was not pH limited. Based on our previous successful use of coating technology for PP and PET [12], extracts from blackberry leaves (*Rubus fruticosus*) and prickly juniper needles (*Juniperus oxycedrus*) from the south-eastern European region, which have not yet been explored for the development a green coating for PLA films, were combined in this work. Since both extracts are extremely potent, combining them with chitosan can be very effective, and their combination is the subject of this investigation.

The specific aims of this study were therefore: (i) to prepare, for the first time, a two-layer application of chitosan extract formulations on the surface of PLA films as a representative of a biodegradable packaging material to pursue a green and sustainable packaging concept; (ii) to evaluate the characteristics of PLA films important for the development of food packaging materials through detailed physicochemical and bioactivity analyses.

## 2. Materials and Methods

### 2.1. Materials

#### 2.1.1. Plant Material

Prickly juniper (*J. oxycedrus*) needles and blackberry (*R. fruticosus*) leaves were harvested at an altitude of 420 m on Kozjak Mountain, Croatia, in August of 2020. The leaves and needles were shade-dried for 7 days at room temperature. The dried material was pulverized and extracted in 50% ethanol using microwave-assisted extraction (MAE) (ETHOS X, Milestone Srl, Sorisole, Italy). The extraction was performed for 5 min at 600 W power. The extracts were filtered, the ethanol was evaporated, and the remaining water was freeze-dried. The composition of individual phenolic compounds of extracts was previously reported in Barbieri et al. [16].

#### 2.1.2. Chemicals and Reagents

Low-molecular weight chitosan (50 to 190 kDa) poly (D-glucosamine), sodium tripolyphosphate (TPP; MW 367.86 g/mol), ethanol (99.8%, GC), acetic acid (≥99.8%; MW 60.05 g/mol), 2,2′-azino-bis (3-ethylbenzothiazoline-6-sulfonic acids) reagent (ABTS), and free radical 2,2-diphenyl-1-picrylhydrazyl (DPPH) were obtained from Sigma-Aldrich (St. Louis, MO, USA). Quaternary chitosan, i.e., Chitosan Quaternary Ammonium Salt, was purchased from CD Bioparticles (Shirley, NY, USA), and the polylactide, polylactic acid—PLA film from Optimont^®^ PLA—Folie, (Bleher Folientehnik GmbH, Germany), the Potassium persulfate from Sigma-Aldrich (99.99%, St. Louis, MO, USA.), Phosphate Buffered Saline (PBS, pH = 7.4) from Chemsolute (Th. Geyer GmbH & Co., Ltd. KG, Renningen, Germany), and Methanol (99.8%) from Honeyweel (Charlotte, NC, USA).

Tryptic soy broth and agar (TSB, TSA; Biolife, Milan, Italy), plate count agar (PCA; VWR BDH Chemicals, Leuven, Belgium), casein peptone, soy peptone, lecithin (Sigma-Aldrich, St. Louis, MO, USA), glycerol, NaCl, Tween 80 (Merck, Darmstadt, Germany) and glucose (Kemika, Zagreb, Croatia) were used for the microbiological analyses.

#### 2.1.3. Solution Preparation

The following liquid formulations in the form of colloidal solutions (macromolecular solutions and dispersions) were used in the study, as shown in Table 1 below.

##### Preparation of Macromolecular Solutions of Primary and Quaternary Chitosan

The primary and quaternary chitosan solutions were prepared in two different concentrations (1 and 2%, *w*/*v*). Deionized water and a few drops of absolute acetic acid (to enable dissolution of the chitosan) were added to a certain mass of chitosan powder. Both solutions were left stirring overnight, and the final volume (100 mL) and pH were adjusted to 4.0 with acetic acid.

##### Preparation of the TPP Solution

Sodium tripolyphosphate (TPP) powder was suspended in deionized water to prepare a 0.2% (*w*/*v*) solution, which was stirred overnight.

##### Preparations of Different Extract Solutions

Dry extracts were suspended in aqueous ethanol (50%; *v*/*v*) to prepare solutions at a concentration of 10 mg/mL.

##### Preparation of Particles with Embedded Extracts

The chitosan particles were prepared using the ionic gelation technique. Simultaneously, 0.2% (*w*/*v*) of TPP solution and different extract solutions, as described above, were added to a fixed volume of 1% (*w*/*v*) chitosan solution (primary and quaternary) to obtain a 5:1 chitosan to TPP weight ratio, chosen according to the methods described in a previously published work [17]. Particles were formed spontaneously under constant stirring for 1 h at room temperature. The final pH of the CHP and QCHP dispersions with different extracts (BBL or JUN) were adjusted to 4.0 with acetic acid.

##### Application of Formulations to PLA Films

All formulations previously prepared with chitosan were applied to the PLA films (size A4: 1682 mm × 2378 mm), which were first cleaned with 70% ethanol and dried. The coating was applied in a roll-to-roll process using a printing table and a magnetic roller (Johannes Zimmer, Kufstein, Austria) at a level 3 rolling speed and level 1 magnet power. The formulations were previously mixed and cannulated onto the films with a fixed volume of 4 mL. The first coating was made of 2% primary or 2% quaternary chitosan, which served for better adhesion of the second layer (chitosan particles with embedded extracts). Under the same conditions, the second layer was formed with primary and quaternary chitosan particles, i.e., CHP and QCHP with different embedded extracts (BBL, JUN). The functionalized PLA films were then air dried. All the prepared samples and their descriptions are shown in Table 2.

### 2.2. Methods

#### 2.2.1. Determination of Zeta Potential (ZP) and Hydrodynamic Diameter (HD)

The zeta potential (ZP) and hydrodynamic diameter (HD) of the prepared particle dispersions were determined on a Litesizer500, Anton Paar (Graz, Austria) particle size analyzer at a temperature of 25 °C. The ZP was measured by electrophoretic light scattering (ELS), which measures the speed of the particles in the presence of an electric field. The HD was measured by dynamic light scattering (DLS). Particles suspended in a liquid are constantly undergoing random motion, and the speed of this motion depends on the size of the particles—smaller particles move faster than larger ones. Before analysis, the dispersion was stirred and adjusted to pH 4 with acetic acid, if necessary. For carrying out the measurements, a diluted sample was poured into an omega cuvette for ZP and size evaluation. The data were collected using Kalliope software (Anton Paar, Graz, Austria).

#### 2.2.2. Goniometry

Static contact angles (SCA) were measured to estimate the surface wettability of the coated formulations using a drop of liquid resting on the surface via a Data Physics Instruments (Fidelstadt, Germany) goniometer. A small drop (5 µL) of Milli-Q water was placed carefully on the surface of the film. A goniometer with SCA 20 software was used to determine CSA at room temperature. Three repetitions were made for each sample.

#### 2.2.3. ATR-FTIR Spectroscopy

The infrared spectroscopy spectra of both chitosans, the pristine PLA film, and functional PLAl films with two-layer coating applications, were measured using a Perkin Elmer Spectrum GX NIR FT-Raman spectrometer (Waltham, MA, USA) equipped with a diamond crystal ATR accessory. To measure the spectra, the background spectra were acquired first, and then the samples were measured under the same conditions as the background, i.e., 16 scans with a resolution of 2 cm^−1^ and in the wavenumber range of 4000–400 cm^−1^. The acquired spectra were deconvoluted using an automatic smoothing filter, automatic baseline corrections, and, finally, ATR corrections. They were normalized to 1 for direct comparison of the different spectra. The same approach was used for analyzing the PLA films after the desorption experiment.

#### 2.2.4. Surface Elemental Composition—XPS Analysis

The chemical composition of the PLA films with the coatings was analyzed with the XPS instrument model TFA-XPS from Physical Electronics (Munich, Germany). The spectrometer was equipped with a hemispherical electron analyzer and monochromatic X-ray source with Al Kα1,2 radiation and the photon energy of 1486.6 eV. The excitation area of the sample was 400 µm^2^. The emitted photoelectrons were measured at a take-off angle of 45°. During the XPS measurements, an electron gun was used for the surface neutralization of the surface charge. The survey spectra were measured at a pass energy of 187 eV and an energy step of 0.4 eV. The software MultiPak v8.1c (Physical Electronics, Munich, Germany) was used for the evaluation of the measured spectra. The same approach was used for analyzing the PLA films after the desorption experiment.

#### 2.2.5. Scanning Electron Microscopy—SEM

To observe the original and two-layer coatings, the PLA films were examined by Scanning Electron Microscopy using a JSM-IT 800SHL instrument (Jeol, Tokyo, Japan). The PLA films were cut and adhered to a double-sided conductive carbon tape, placed on a holder, and sputtered with gold to ensure conductivity and prevent charging effects. The samples were examined with an accelerating voltage of 5 kV, along with a variable working distance and comparable magnification. The images were acquired using a secondary electron detector.

#### 2.2.6. Desorption Study

Desorption of the functionalized films was carried out by weighing 0.5 g of the functionalized film, cutting it into small pieces, and soaking it in 25 mL of Milli-Q water. The beakers were placed on a shaker for 24 h. The removed film pieces (after desorption) were dried overnight at room temperature and prepared for ATR-FTIR and XPS analysis, as described above. These two methods were used to determine the migration of elements and compounds of chitosan and phenolics from the plant extracts directly from the films, and to analyze the residual components/elemental composition of the films. The desorption was then evaluated by comparing the results before and after desorption.

#### 2.2.7. Antioxidative Activity

Antioxidative activity was determined using ABTS and DPPH assays. Both assays were based on the spectrophotometric determination (UV-VIS) of the decolorization of the reagent in the presence of an antioxidant. The results obtained by both assays were further compared.

##### ABTS Assay

A 7 mM ABTS^•+^ solution was prepared in 2.45 mM potassium persulfate. Before starting the analysis, the ABTS^•+^ solution was diluted with PBS buffer to reach an absorbance value of 0.7161 at 734 nm. Then, 3.9 mL of the ABTS^•+^ solution was added to 0.1 g of film. The inhibition was estimated spectroscopically by measuring the absorbance at 734 nm at 25 °C, at the time points 0 min, 15 min, and 60 min. The results are presented in percentages of inhibition and were calculated using Equation (1):
Inhibition (%) = (A_Control_ − A_Sample_)/A_Control_ × 100(1)
where A_Control_ is the absorbance measured at the starting concentration of the reagent solution, and A_Sample_ is the absorbance of the remaining concentration of the reagent solution in the presence of the functionalized film [18].

##### DPPH Assay

First, 0.081 mM DPPH^•^ solution was prepared in absolute methanol. Pieces of the films were cut into 1 cm × 1 cm squares and then covered with 1 mL of DPPH^•^ solution with an initial absorbance of 1.0904. The absorbance was measured at a wavelength of 517 nm at the time points 0 min, 15 min, and 60 min. The experiment was performed twice. The results are presented in percentages of inhibition and were calculated using Equation (1).

#### 2.2.8. Antimicrobial Activity

To test the antimicrobial activity of the functionalized PLA films, *Staphylococcus aureus* ATCC 25923 was selected as a model microorganism. *S. aureus* was stored in TSB with glycerol at −80 °C in the culture collection of the Laboratory for Food Microbiology at the Department of Food Science and Technology, Biotechnical Faculty, University of Ljubljana, Slovenia, and was revived on a TSA at 37 °C for 24 h. The inoculum was prepared and inoculated on films for evaluation using a modified ISO 22196 (2007), [19] method for measuring antimicrobial activity on plastic surfaces. After 24 h incubation at 37 °C, the viable bacteria count per cm^2^ was determined with PCA, using the pour plate method, and the antimicrobial activity (R) and percentage of antimicrobial activity (R (%)) were calculated using Equations (2) and (3), respectively:(2)$$R=U_{t}-A_{t}$$
(3)$$R\left(\%\right)=\left(1-10^{-R}\right)\times 100$$
where U_t_ is the average of the logarithm of the number of viable bacteria, in cells/cm^2^, recovered from the control films after 24 h incubation, and A_t_ is the average of the logarithm of the number of viable bacteria, in cells/cm^2^, recovered from the functionalized films after 24 h incubation. Where there was no visible bacterial growth on the plates, the volume of dilution was measured to calculate the number of viable bacteria (cells/cm^2^), and the results are presented as ‘smaller than this value.’

#### 2.2.9. Statistical Analysis

The results for contact angle, as well as antioxidant (ABTS, DPPH) and antimicrobial activity, are presented as mean ± standard deviation. Data were normally distributed (Shapiro–Wilk test, *p* > 0.05). The results were compared using analysis of variance and a post hoc Duncan test. A statistical significance of *p* < 0.05 was considered for all tests. The analysis was performed with the IBM SPSS Statistics 25 program.

## 3. Results and Discussion

### 3.1. Particle Size and Zeta Potential Determination

The hydrodynamic diameter (HD) and zeta potential (ZP) for all liquid formulations (chitosan particles alone, or with embedded extract) used for PLA film functionalization are given in Table 3. The pH of all formulations was calibrated to 4.0.

It was predicted that the particles with incorporated/coated extracts would have a larger average size in terms of the intensity of the synthesized particles, and this was shown to be the case. The average particle size for the CHP was around 358 nm, and for the QCHP it was approximately 240 nm, while, for both chitosan particles with embedded extracts, it ranged from 8980 to 20,402 nm. For both extracts used, the increase in particle size was more pronounced in the CHP compared to the QCHP. It was evident from these sub facts that the incorporation of the extracts into or on the surface of the chitosan particles increased the hydrodynamic diameter from a colloidal size to a micron scale, therefore, we no longer refer to colloidal dispersion, but rather to particle suspension. However, the increase in particle size confirmed the successful incorporation of the extracts and binding of these active substances to the interior and/or surface of the chitosan particles [15,18]. The PDI was generally not high and showed a fairly uniform size distribution of the particles. It can also be seen that the PDIs of the chitosan particles with embedded extract were higher than those of the chitosan particles themselves, which was also indirect evidence of the successful inclusion of the extract in the chitosan particle matrix, thus showing a broader size distribution.

The Zeta potential measurements showed that the formulation with chitosan particles was stable at a pH of 4.0, which was due to the presence of protonated amino groups on the surface of the particles. According to the literature [18], dispersions/suspensions with a zeta potential higher than ±30 mV are defined as stable, with minimal sedimentation tendency. Both CHP and QCHP achieved this state, but only one particle-based formulation with embedded extract, namely QCHP’sJUN, showed a zeta potential above 30 mV, with a zeta potential of almost 40 mV, indicating particle stability. In contrast, lower zeta potential values were achieved for other formulations, with lower values in CHP. The lowest zeta potential value was found in the CHP’sBBL formulation, and it accounted for 12 mV. This could indicate a tendency for particle agglomeration, as fewer protonated amino groups were available. In the case of embedded extracts, the accessibility of the amino groups of chitosan was lower, and therefore, the zeta potentials of the extract formulations were also slightly lower compared to that for chitosan alone. This could indicate that the extracts were also attached to the surface of the chitosan particles, or that the phenolic compounds may bind to the amino groups chemically, thereby reducing the amount of available (free) amino groups, thus lowering the positive zeta potential [15,18]. The formulations of the extracts in combination with quaternary chitosan (as a particle suspension) achieved higher zeta potential values, on average, than the formulations of the extracts in combination with primary chitosan, indicating their stability compared to the instability of the primary chitosan particles. This may be due to the higher positive charge density of the quaternary chitosan, or to the fact that the quaternary group did not allow binding to the extracts due to steric hindrances.

### 3.2. Goniometry

Evaluating the wettability of a packaging film is extremely important for practical applications. The reduction of the contact angle and the conversion of the surfaces to hydrophilic forms reduces the potential condensation process of the film within the packaging system. Condensation in contact with food, in turn, deteriorates the packaging conditions and increases the potential for food spoilage and contamination. It also negatively influences film transparency [20]. The contact angles of PLA and functionalized PLA packaging films are given in Table 4.

The PLA reference film had an average contact angle of 77.56°, demonstrating the hydrophilicity of the film, but with a rather high contact angle. In macromolecular chitosan solutions alone, it can be seen that the first layer increased the contact angle of the reference PLA slightly, but it still remained hydrophilic. The functionalization of the films with an additional second layer coating of chitosan with embedded extracts decreased the contact angle for all samples, increasing their hydrophilicity. The greatest impact for both extracts was shown in samples where quaternary chitosan was used for both functionalization layers of the PLA film, and the lowest SCA was reached by the PLA + 2QCH + QCHP’sBBL sample (Table 4). It has been shown that the application of the developed bilayer coatings on the surface of the PLA films improves hydrophilicity and preserves condensation without affecting transparency, which is of great value for practical applications in the food industry. In all cases, the surface of the bilayer coated PLA films became significantly more hydrophilic, initiating anti-fogging properties, contributing to a delay in food spoilage [17].

### 3.3. ATR-FTIR Spectroscopy

Using infrared spectroscopy, we aimed to verify the presence of functional chemical groups of chitosan and the extracts and to evaluate the success of their application to PLA film, first in the form of a macromolecular colloidal dispersion, and then in the form of chitosan’s embedded with different extracts. Thus, chitosan (primary, quaternary), plant extract (JUN and BBL), and 8 functionalized films were analyzed (Figure 1).

Figure 1a shows the infrared spectrum of pure primary (CH) and quaternary chitosan (QCH) as refences for chitosan fingerprints. The band in the 3283 cm^−1^ region corresponds to the amine functional group N-H, hydroxyl group O-H, and intramolecular hydrogen bonds. The signal in the 2870 cm^−1^ region can be attributed to the molecular vibrations of the C-H bond. These described bands are typical characteristic bands of carbohydrates. The presence of the remaining N-acetyl groups can be confirmed by the following peaks: peak 1652 cm^−1^ corresponds to the C=O functional group in primary amide I, while peak 1376 cm^−1^ corresponds to the C-N group in the secondary amide. The band at 1147 cm^−1^ can be attributed to the vibrations of the C-O-C bridge, while the band at wavenumber 1029 cm^−1^ corresponds to the vibrations of the C-O bond [15,17].

In general, a wide range of derivatives can be obtained from CH using chemical modification. One of these is also quaternary chitosan, which is obtained by intensive methylation of the primary amino group of chitosan [21]. As in the case of CH, we observed a characteristic band in the region of 3247 cm^−1^ corresponding to the N-H and O-H functional groups (Figure 1a). The major difference between the two spectra was in the region of 1478, 3010 and 960 cm^−1^, where new signals were observed that were not present in the primary chitosan. These can be attributed to vibrations in N^+^-(CH_3_)_3_ [22]. These peaks most likely correspond to the asymmetric angular bending of the quaternary CH_3_ hydrogen methyl group that occurs during the methylation of chitosan [22,23]. The absorption peak at wavenumber 1029 cm^−1^, which is consistent with that of the primary chitosan, corresponds to the C-O bonds.

In the case of the pure extracts (Figure 1b,c; spectra highlighted in red), namely JUN and BBL, all spectra showed similar bands at comparable wavenumbers in the same wavelength ranges. The first observed absorption peak was in the range of 3311–3336 cm^−1^ and was associated with the presence of the hydroxyl groups -OH, characteristic of phenolic compounds. The next peak was the same for both extracts, and was located at about 1637 cm^−1^. This peak can be attributed to the presence of several functional groups: the C=C group, vibrations in aromatic rings, vibrations of the amine group N-H, group C=O in the amides, or the presence of carboxyl groups. However, this absorption band can also be associated with the presence of flavonoids and amino groups. The last band observed in the spectra was at 1075 cm^−1,^, which can be associated with the presence of secondary alcohols and/or vibrations of the C-O ether functional group [24,25,26]. Phenolic compounds of JUN and BBL extracts were identified in our previous study. The JUN extract contained high amounts of vanillic acid (10.51 mg/L), and apigenin, rutin, and catechin were determined as the dominant flavonoids, found in quantities of 7.66, 6.95, and 4.86 mg/L, respectively. In the BBL extract, rutin (a flavonoid), and chlorogenic acid were found to be dominant in quantities of 29.88 and 6.22 mg/L, respectively [16]. Among other phenolic compounds in the JUN extract, gallic, protocatechuic, and *p*-hydroxybenzoic acids were detected, as well as epicatechin and epigallocatechin gallate, while gallic, caffeic, and *p*-coumaric acid, along with epicatechin and astringin, were found in the BBL extract.

The spectra in Figure 1b and c also show the pure FTIR spectrum of the native PLA film. The unmodified PLA film showed typical bands between 3600–3000 cm^−1^ corresponding to the O-H groups. The band at 1750 cm^−1^ corresponded to the vibrations of the C=O functional group, while the wavenumbers at 1180 and 1084 cm^−1^ were attributed to the vibrations of the C-O-C bridge [27,28].

After applying the two-layer coatings on PLA films, some differences could be observed in the spectra of the differently functionalized PLA films. For the PLA films modified with different formulations that also contained particles with JUN extract (Figure 1b), the most obvious difference could be observed for the sample PLA + 2%CH + QCHP’sJUN, where the appearance of the N-H and O-H peaks occurred at larger wavenumbers. In addition, some small peaks at 1640 cm^−1^ could be observed for the foils in Figure 1b, which were typical of JUN extract. Nevertheless, all the spectra in Figure 1b were the same as those in the PLA film. However, some functional groups also overlapped at the same wavenumber, making it difficult to distinguish the bands for both chitosan’s. For the PLA with chitosan applied and particles with BBL extract (Figure 1c), a similar conclusion can be drawn, as in the case of Figure 1b. However, the difference was again seen at larger wavenumbers, where the intensity of the peaks typical for the N-H and O-H groups decreased in the following manner: PLA + 2%CH + QCHP’sBLL, PLA + 2%QCH + CHP’sBBL and PLA + 2QCH + QCHP’sBBL samples. Overall, the infrared spectroscopy clearly indicated the successful application of the two-layer coating on the PLA film.

### 3.4. ATR-FTIR Spectroscopy after Desorption

The ATR-FTIR spectra of PLA films after being subjected to desorption are shown in Figure 2.

Exemplary selected films were exposed to the desorption bath, and the spectra of films after being subjected to the desorption were recorded and compared with the original functionalized films, with PLA as the reference film (Figure 2). It was qualitatively determined that no large specific changes were detected after desorption, indicating partial coating stability. Small changes were observed at larger wavenumbers for the sample PLA + 2%QCH + QCHP’sJUN, where the intensity of the bands corresponding to the functional groups—OH and -NH decreased (Figure 2a,b), while this was not the case for sample PLA + 2%CH + CHP’sBBL (Figure 2b). In all, two measured spectra, the signal at 1082 cm^−1^, as well as at 1750 cm^−1^, decreased after desorption, which is typical for C=O. All these decreases suggest that chitosan and extract as a combined coating migrated, to some extent, from the surfaces of the foils.

### 3.5. XPS

The XPS chemical composition of the PLA films with JUN extract, before and after desorption, is shown in Table 5. For the uncoated PLA film as the reference, 66.4, 32.9, and 0.7 at % of carbon, oxygen and silicon were found, respectively. When comparing the composition of the PLA films with the JUN extract with the uncoated film, we can observe a lower oxygen content and the presence of several additional elements in minor concentrations. Among these, the most abundant element was nitrogen (2.7–4.9 at. %) belonging to the chitosan, whereas other elements—P, S, Na, Cl, and Si—were present in concentrations far below 1%. The results, summarized in Table 5, confirmed the presence of chitosan with JUN extract on the surface of the PLA films. Although chitosan is rich in oxygen, JUN extract is almost completely lacking in oxygen, which explains the overall decrease in oxygen concentration in comparison with the uncoated film. The other minor elements—Mg, Cl, Ca, S, and K—belonged to the plant extract, as these elements were not found in the chemical structure of chitosan. These elements were derived from plant minerals. For example, the mineral constituents (Ca, Na, K, Mg, Cu, Zn, Fe, and Mn) were detected in extracts from Juniperus phoenicea seeds and analyzed in a study by Nasri et al. using an atomic absorption photometer [29].

The presence of sodium and phosphorus can also be explained as originating from the chitosan particles, as they were synthesized by the addition of sodium triphosphate. After desorption, the oxygen concentration increased, becoming similar to the uncoated PLA. Furthermore, the nitrogen concentration decreased, and the presence of the other minor elements decreased, and in some cases, were no longer noticeable. From these results, we can conclude that part of the two-layered coating was removed from the surface after desorption.

The second extract used in this investigation was the BBL extract, and the XPS results for the PLA films with BBL extract are shown in Table 6. Here, we can also clearly observe the presence of nitrogen and several other elements on the surface—as in the previous case—thus proving the successful application of the coating. As mentioned earlier, other elements, such as Na, Mg, Si, P, Cl, and S, may be derived from the extract minerals. Staszowska-Karkut and Materska (2020) reported on the mineral content in berry plant extracts, and found that minerals such as potassium (K), calcium (Ca), magnesium (Mg), phosphorus (P), sodium (Na), iron (Fe), copper (Cu), zinc (Zn), manganese (Mn), and boron (B) are present in leaf extracts [30].

Within the detection limits of the XPS spectrometer, the composition of the microelements was not changed after desorption. However, this time, it was noted that nitrogen was increased in all samples after desorption. As explained before, this can be explained by the nonuniformity of the coatings; the removal of some remains of the by-products from the applied coating, or the removal of some extracts not embedded in the chitosan particles, can be seen from SEM (as discussed below), thus revealing the chitosan particles underneath.

### 3.6. SEM Analysis

SEM analysis was also performed, the results of which are presented in Figure 3 and in the commentary below.

To further investigate the morphology of the PLA films after two-layer coating, SEM was used to visualize the coverage of the films (Figure 3). The pristine PLA film showed a typical flat surface with some impurities, possibly due to contamination, as also shown with XPS (Si). The coated samples showed a completely different morphology, which can be clearly seen in the adhesion and roughness of the two applied layers. In the case of CHP and QCH with JUN extract (Figure 3, images labelled PLA + 2%CH + CHP’sJUN and PLA + 2%CH + QCHP’s JUN), compared to the untreated PLA, lower magnification showed complete coverage of the PLA film with clearly visible particle agglomerates, and some particles were about 100 nm in size (see the higher magnification in the insets of PLA + 2%CH + CHP’s JUN and PLA + 2%CH + QCHP’s JUN). Samples with BBL extracts showed uniform coverage in the first applied layer, while the second application of both types of Ps with BBL extracts was distributed randomly in the form of agglomerates. However, the larger magnification in Figure 3 shows the presence of particles with sizes larger than 100 nm. For the first 2% QCH applications, i.e., samples PLA + 2%QCH + CHP’s JUN, PLA + 2%QCH + QCHP’s JUN, PLA + 2%QCH + CHP’BBL, and PLA + 2%QCH + QCHP’sBBL, some similar morphologies were observed, as for the samples with the first 2% coating of CH. Nevertheless, some differences could be seen, due to the presence of fewer agglomerates and uniformly distributed NPs. This was especially true for sample PLA + 2%QCH + QCHP’s BBL. Samples PLA +2 %QCH + CHP’s JUN, PLA + 2%QCH + QCHP’s JUN, and PLA + 2%QCH + CHP’s BBL showed uneven coverage, with larger agglomerates consisting of smaller CH, or QCHP with embedded extracts. In most samples, we could see that the coatings were not homogeneous. The particle size showed lower values compared to the determination of the hydrodynamic diameter, but we must consider that here, the particles were in the dry state, while the hydrodynamic radius includes all water molecules and clouds attracted by the particles. However, a small population of particles (Gauss distribution, not shown here) of around 100 nm in size was also detected by Litesizer.

### 3.7. Antioxidant Efficiency of Functionalised PLA Films

The antioxidant activity test was performed to determine the effect of coatings on the surface of PLA films on inhibiting the oxidation process that often occurs during the storage of food. Two methods were used for this purpose—ABTS assay and DPPH assay. The results are shown in Figure 4 and Figure 5, respectively. The percentage of inhibition was determined at three time points (0 min, 15 min, and 60 min). As can be seen in Figure 4a and Figure 5a, CH and QCH had no antioxidant potential.

In functionalized films, the observation of the kinetics shows that the antioxidant activity increases with time, indicating that natural substances require a certain time for their activity, which is related to the following chemical phenomena (protonation and swelling, conformational changes due to solubility, extract release, etc.) [18]. In functionalized PLA films with BBL and JUN extracts, the results were comparable between films. With the ABTS assay (Figure 4), 100% inhibition was determined after 60 min for all samples except PLA + 2%QCHP + CHP’sBBL and PLA + 2%QCHP + QCHP’sJUN, which was probably due to non-homogeneous coating on the PLA films, as was confirmed by SEM. Moreover, in the DPPH assay, the highest activities were determined after 60 min, and comparable inhibition trends were reached between the samples, with the best results in PLA + 2%QCH + CHP’sBBL and PLA + 2%QCH + CHP’sJUN. Moreover, it may also be pointed out that as coatings, chitosans themselves express negligible antioxidant activity, and thus, it may be concluded that extracts are the main driving force for free radical inhibition.

Based on the results of both methods, similar trends, but lower values, were obtained for the DPPH method due to some differences in methods. It has already been discussed that ABTS radical cation (ABTS^•+^) is reactive towards most antioxidants, and it is soluble in both aqueous and organic solvents. The ABTS^•+^ method is a useful tool in determining the antioxidant activity of both lipophilic and hydrophilic antioxidants in various matrices (body fluids, food extracts, etc.) [31]. ABTS^•+^ reacts rapidly with antioxidants, and they can be applied over a wide pH range. The selected substances, including most phenolic compounds, reduce ABTS^•+^ if its redox potential is lower than that of the ABTS (0.68 V). The DPPH assay was based mainly on the electron transfer reaction, while hydrogen—atom abstraction is a marginal reaction pathway [32]. The interactions between antioxidants and DPPH^•^ were also determined by the antioxidant’s structural conformation. Some compounds react very rapidly with DPPH^•^, and they reduce the number of DPPH^•^ molecules corresponding to the number of available hydroxyl groups. Nevertheless, this mechanism seems to be more complex, and the observed reactions are slower in most antioxidants [33]. The DPPH method yielded lower antioxidant values than the ABTS assay, probably because the DPPH method has more limitations. As shown above, the DPPH method was characterized by a lower sensitivity than the ABTS assay. The reaction of DPPH^•^ with most antioxidants is slower than in the case of ABTS^•+^. Moreover, DPPH dissolves only in polar matrices, and an extract may contain components whose spectra overlap with the DPPH^•^ spectra, which can significantly distort the spectrophotometric measurement [33]. In our case, it can be concluded that, due to the hydrophilic coatings and aqueous solvents, the ABTS assay was a more reliable and trustworthy method to monitor antioxidant activity.

### 3.8. Antimicrobial Activity

The antimicrobial activity of the functionalized PLA films against the Gram-positive bacteria *S. aureus* is presented in Figure 6 and Table 7. The highest antimicrobial activity was found for the reference PLA films with chitosans, where no culturable cells were determined. All the functionalized PLA films showed antimicrobial activity, but with large standard deviations, which was probably the result of the non-homogeneous coating on the PLA films, as confirmed by the SEM analysis. However, antimicrobial activities were assessed with 1.4 to 3.7 log CFU/cm^2^ in the PLA films with BBL, and 3.3 to 4.6 log CFU/cm^2^ in PLA films with JUN, which represents a 95.6 to 99.9% reduction in the number of *S. aureus* bacteria on the films. PLA as a reference foil, however, showed no inhibiting effect on the tested bacteria.

In the case of chitosan, it was primarily the protonated amino groups that exerted antimicrobial activity. It has been shown that higher amounts of available protonated amino groups indicate an increased bacterial reduction [34]. Concerning extracts, it has been reported that large numbers of different chemical compounds, such as phenolic compounds and their derivative compounds, the esters of weak acids, fatty acids, terpenes, and others, are present in different ethanolic extracts, and these chemical components can affect multiple target sites on the bacterial cells [35,36]. With the combination of the included extract (juniper and blackberry) as second layer PLA coatings, CHP extracts and QCHP extract suspensions, also showed great antimicrobial activity, and, in combination with chitosan, did not lessen its function significantly. The addition of these extracts to the coating of PLA films reduced antimicrobial activity by up to 5%. A negligible lower percentage of antimicrobial activity may be attributable to the reduced availability of the amine groups of chitosan. It is well known that, among all inhibitory mechanisms, a positive charge has a significant and usually dominant effect on antimicrobial activity, which is due to electrostatic interaction with the anionically charged bacterial cell wall and consequently, its collapse and destruction [34].

## 4. Conclusions

This paper presents the functionalization of PLA using our previously successfully developed technology allowing for the functional coating of chitosan particles with embedded polyphenols for potentially active packaging, which works on both PP and PET. We have focused mainly on commercially used phenols/polyphenols, such as catechin and resveratrol, but here, we focus on two plant extracts (blackberry leaves and juniper needles) derived from the biomass of our southeastern European region. They were embedded in chitosan biopolymer particles and dispersed in a macromolecular chitosan solution, then deposited in two layers on PLA films.

The ATR-FTIR and XPS spectra confirmed that both layers were bound successfully to the film’s surface, with some desorption. In addition, the hydrophilicity of the films was reduced by decreasing the contact angle, i.e., its wettability was improved. The latter is very important for ensuring food safety and quality due to its anti-fogging effect. The images on SEM showed predominantly non-uniform coatings, which was to be expected when using a suspension of particles dispersed in a macromolecular solution with a ZP below +/− 30 mv (lack of stability and agglomeration tendency). Using the ABTS method, which was found to be more reliable compared to the DPPH method, an increase in antioxidant activity in the range of 83.50–100% was observed for all functionalized bilayer films.

The films prepared by these methods showed a strong reduction in *Staphylococcus aureus* by more than 95% for all functionalized films.

Although the chitosan coatings themselves showed great antimicrobial activity, the second layer, with its combination of phenolic extracts of juniper or blackberry, was very important when considering antioxidant activity, for which chitosan showed no activity, but in combination with the extract, the films showed great antioxidant activity. The same was true for hydrophilicity. Therefore, it is necessary to evaluate all these properties together to assess their synergy, along with their real biological value and potential for food applications. The only drawback of this technology is the inhomogeneity, which requires further optimization steps, such as using extracts as a second layer alone rather than embedded in particles, or changing the particle synthesis route, which is already in progress.

The presented method, with some optimization steps for the development and application of PLA coatings, is suitable for various applications where such film properties are desired.

Examples of applications include the packaging of food products such as meat, vegetables, dairy and bakery products, pharmaceuticals, etc.

In addition, the concept is environmentally friendly, as the active functional compounds can be recovered from biomass and reused to develop value-added products, including active packaging. In addition, the biodegradability of PLA is not expected to deteriorate, but may even improve, as the integrated active ingredients are of natural and bio-based origin.

## Figures and Tables

**Figure 1 jfb-13-00228-f001:**
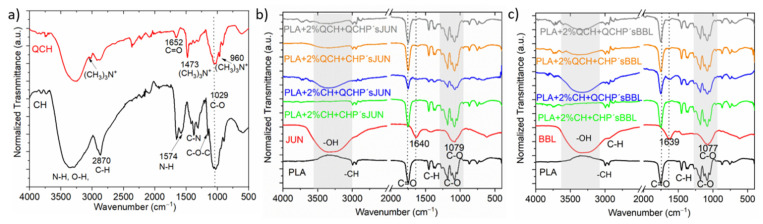
ATR-FTIR spectra of CH and QCH (**a**), and of pure PLA and the extracts (JUN, BBL), along with all functionalized foils (**b**,**c**).

**Figure 2 jfb-13-00228-f002:**
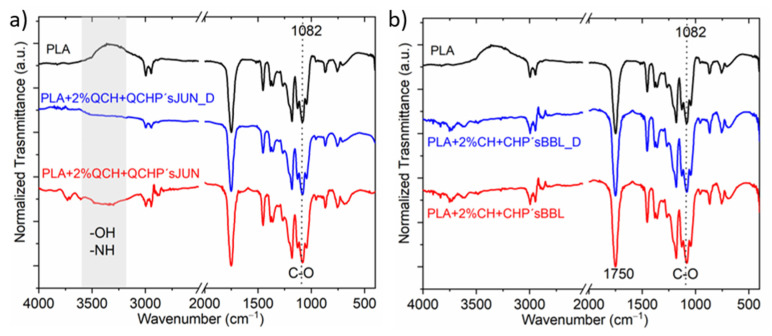
Example of selected films with different applications, before and after the desorption process; (**a**) PLA + 2%QCH + QCHP’s JUN before and PLA + 2%QCH + QCHP’s JUN _D after desorption, and (**b**) Sample PLA + 2%CH + CHP’s BBL before and after PLA + 2%CH + CHP’s BBL _D desorption.

**Figure 3 jfb-13-00228-f003:**
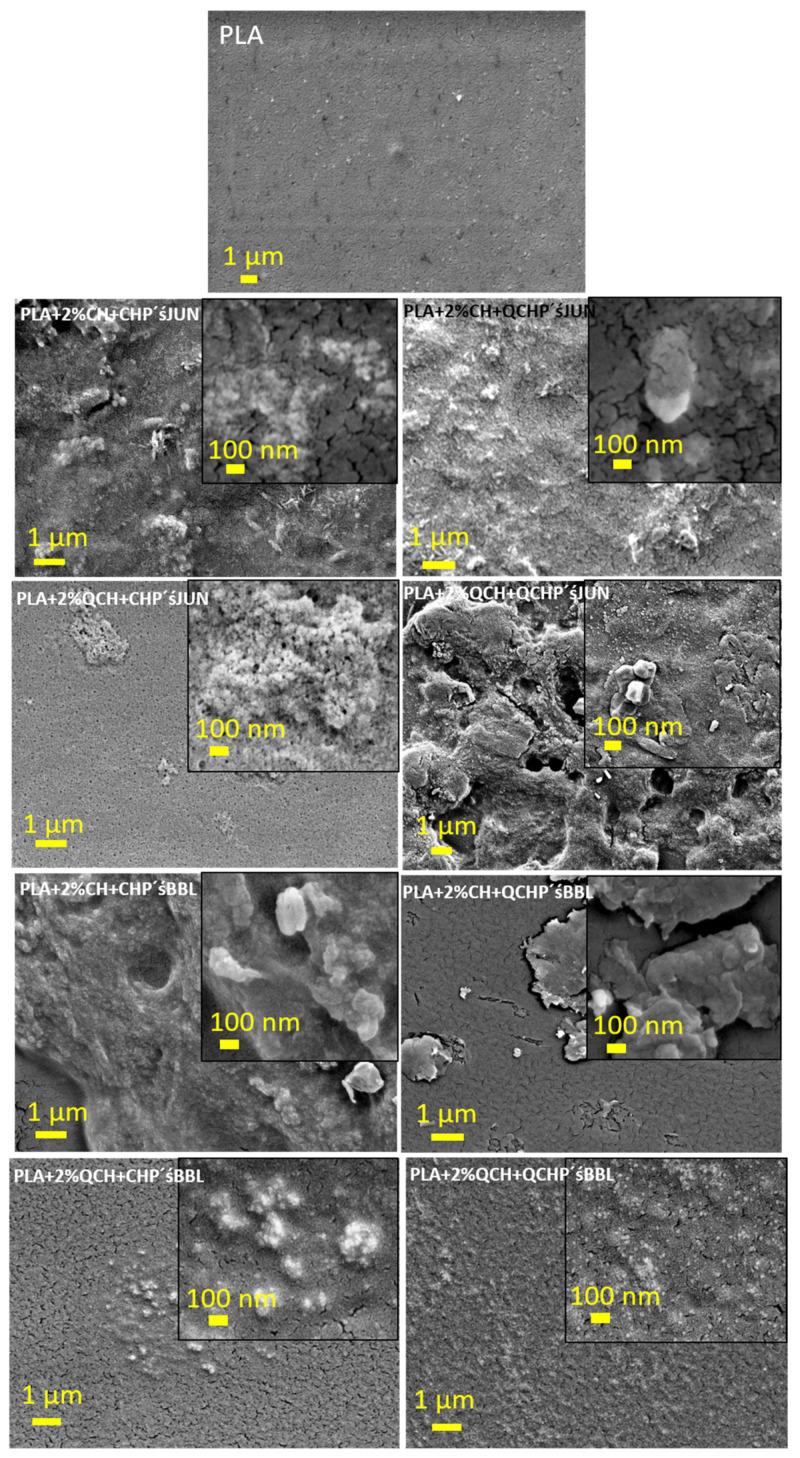
SEM images for PLA films coated with first 2% CH or QCH, followed by a second layer in the form of micro/nanoparticles of CH, or QCH with embedded extracts (namely JUN and BBL).

**Figure 4 jfb-13-00228-f004:**
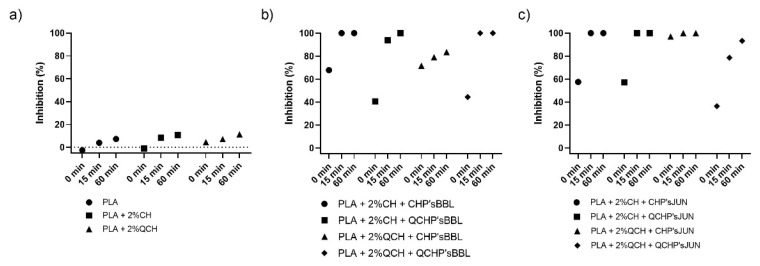
Percentage of inhibition of oxidation of the reference PLA films (**a**) and functionalized PLA films ((**b**),BBL extract; (**c**), JUN extract) determined with an ABTS assay.

**Figure 5 jfb-13-00228-f005:**
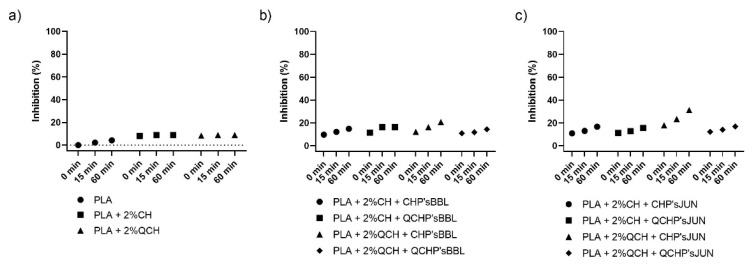
Percentage of inhibition of oxidation of the reference PLA films (**a**) and functionalized PLA films ((**b**), BBL extract; (**c**), JUN extract) determined with a DPPH assay.

**Figure 6 jfb-13-00228-f006:**
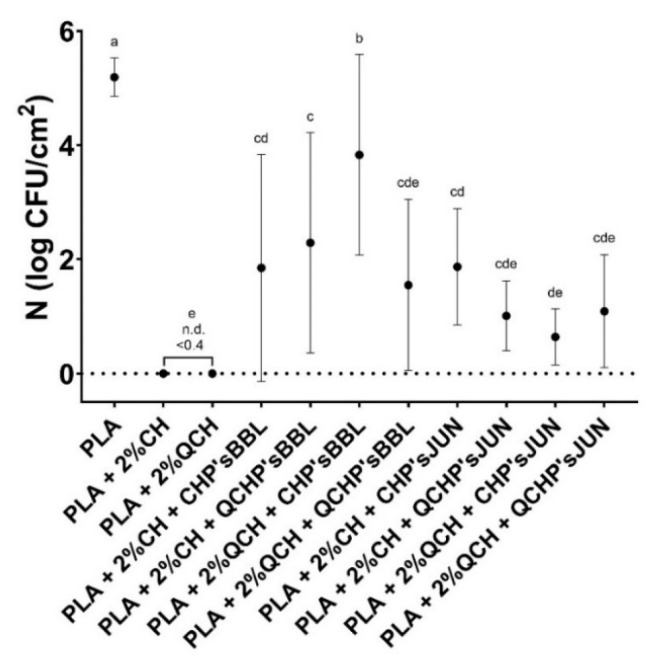
Number of *Staphylococcus aureus* determined on the reference PLA films and functionalized PLA films after 24 h. There was no statistically significant difference between the average values of the samples marked with the same letter.

**Table 1 jfb-13-00228-t001:** Sample description of the prepared solutions.

Solutions	Acronym
1% Primary chitosan	CH
1% Quaternary chitosan	QCH
2% Primary chitosan	2% CH
2% Quaternary chitosan	2% QCH
Primary chitosan particles	CHP
Quaternary chitosan particles	QCHP
Blackberry leaves extract	BBL
Juniper needles extract	JUN
Sodium tripolyphosphate	TPP

**Table 2 jfb-13-00228-t002:** Sample descriptions.

Description of the Samples	Sample Notation
PLA with no coatings	PLA
PLA applicate with 2% CH	PLA + 2% CH
PLA applicate with 2% QCH	PLA + 2% QCH
**Samples with primary chitosan as first layer**	
PLA applicate with 2% CH and CHP’sJUN	PLA + 2% CH + CHP’JUN
PLA applicate with 2% CH and QCHP’sJUN	PLA +2% CH + QCHP’s JUN
PLA applicate with 2% CH and CHP’sBBL	PLA + 2% CH + CHP’sBBL
PLA applicate with 2% CH and QCHP’sBBL	PLA + 2% CH + QCHP’sBLL
**Samples with quaternary chitosan as first layer**	
PLA applicate with 2% QCH and CHP’sJUN	PLA + 2% QCH + CHP’sJUN
PLA applicate with 2% QCH and QCHP’sJUN	PLA + 2% QCH + QCHP’sJUN
PLA applicate with 2% QCH and CHP’sBBL	PLA + 2% QCH + CHP’sBBL
PLA applicate with 2% QCH and QCHP’sBBL	PLA + 2% QCH + QCHP’sBBL

**Table 3 jfb-13-00228-t003:** HD with polydispersity index (PDI) and ZP with standard deviation (SD) of zeta potential of different formulations.

Sample	HD [nm]	PDI [%]	ZP [mV]	SD
CHP	358	16.8	36	0.2
QCHP	239.9	17.5	33	0.3
CHP’sBBL	20,402	39.5	12	0.2
QCHP’sBBL	14,201	27.2	24.8	0.3
CHP’sJUN	16,264	23.7	14.9	0.4
QCHP’sJUN	8980	29.4	39.8	0.9

**Table 4 jfb-13-00228-t004:** Static contact angles (SCA) for PLA reference samples and functionalized films as mean ± standard deviation and calculated difference in percentages, compared to the PLA reference film. There was no statistically significant difference between the average values of the samples marked with the same letter.

Samples	SCA (α/°)	Difference (%)
PLA	77.56 ± 1.61 ^a^	/
PLA + 2% CH	80.39 ± 2.23 ^a^	−3.66 ± 2.23
PLA + 2% QCH	78.69 ± 4.02 ^a^	−1.47± 4.02
PLA + 2% CH + CHP’sBBL	40.00 ± 0.53 ^de^	48.43 ± 0.53
PLA+2% CH+QCHP’sBBL	42.66 ± 2.09 ^cd^	44.99 ± 2.09
PLA + 2% QCH + CHP’sBBL	44.96 ± 0.24 ^c^	42.03 ± 0.24
PLA + 2QCH + QCHP’sBBL	30.33 ± 2.49 ^f^	60.90 ± 2.49
PLA + 2% CH + CHP’sJUN	50.41 ± 1.92 ^b^	35.00 ± 1.92
PLA + 2% CH + QCHP’sJUN	38.84 ± 3.34 ^de^	49.92 ± 3.34
PLA + 2% QCH + CHP’sJUN	37.67 ± 2.30 ^e^	51.43 ± 2.30
PLA + 2% QCH + QCHP’sJUN	37.61 ± 1.34 ^e^	51.51 ± 1.34

**Table 5 jfb-13-00228-t005:** XPS analysis of functionalized PLA films with juniper needle extract, before and after desorption (at. %).

Sample	C	N	O	Na	Mg	Si	P	Cl	Ca	S	K
PLA + 2% CH + CHP’sJUN	65.9	3.4	27.1	/		3.2	0.2	0.3			
PLA + 2% CH + QCHP’sJUN	65.8	4.9	27.8	/		0.8	0.5			0.2	
PLA + 2% QCH + CHP’sJUN	65.2	3.8	28.8	0.6		0.6	0.3	0.3		0.5	
PLA + 2% QCH + QCHP’sJUN	66.8	2.7	29.7			0.3	0.2			0.3	
**After Desorption**											
PLA + 2% CH + CHP’sJUN	62.8	1.5	34.4	1				0.2			
PLA + 2% CH + QCHP’sJUN	63.8	3	31.4	0.9			0.3	0.6			
PLA + 2% QCH + CHP’sJUN	63.1		34.2	1.5			0.2	0.3	0.1		0.5
PLA + 2% QCH + QCHP’sJUN	63.5	2.1	32.6	0.7			0.4	0.6			
**Difference**											
PLA + 2% CH + CHP’sJUN	−3.1	−1.9	7.3	−1			0	−0.1			
PLA + 2% CH + QCHP’sJUN	−2	−1.9	3.6	−0.9			−0.2	0.6			
PLA + 2% QCH + CHP’sJUN	−2.1	−3.8	5.4	0.9			−0.1	0			
PLA + 2% QCH + QCHP’sJUN	−3.3	−0.6	2.9	0.7			0.2	0.6			

**Table 6 jfb-13-00228-t006:** XPS Analysis of functionalized PLA films with the BBL extract, before and after desorption (at. %).

Sample	C	N	O	Na	Mg	Si	P	Cl	Ca	S	K
PLA + 2%CH + CHP’sBBL	62.9	2.1	33.2	0.4	0.3	0.2	0.6	0.3			
PLA + 2%CH + QCHP’sBLL	64.4	2.2	31	1		0.3		1			
PLA + 2%QCH + CHP’sBBL	66.7	2.8	29			0.7	0.6	0.2			
PLA + 2QCH + QCHP’sBBL	66.4	2.1	28.9	0.5		1.1		0.9			
**After Desorption**											
PLA + 2%CH + CHP’sBBL	62.6	4.4	30.6	0.4		1.1	0.6	0.2		0.1	
PLA + 2%CH + QCHP’sBLL	64	4.5	30.2			0.6	0.4			0.3	
PLA + 2%QCH + CHP’sBBL	67.1	3.6	26.9			1.6		0.6		0.2	
PLA + 2QCH + QCHP’sBBL	67	4.9	26.9			0.3	0.2	0.5		0.2	
**Difference**											
PLA + 2%CH + CHP’sBBL	−0.3	2.3	−2.6	0	0.3	0.9	0	−0.1			
PLA + 2%CH + QCHP’sBLL	−0.4	2.3	−0.8	−1		0.3	0.4	−1			
PLA + 2%QCH + CHP’sBBL	0.4	0.8	−2.1	0		0.9	−0.6	0.4			
PLA + 2QCH + QCHP’sBBL	0.6	2.8	−2	−0.5		−0.8	0.2	−0.4			

**Table 7 jfb-13-00228-t007:** Antimicrobial activity of the reference PLA films and functionalized PLA films against the Gram-positive bacteria *Staphylococcus aureus*, presented as logarithmic reduction (R) and percentage of reduction (R (%).

Sample	Bacteria	
	*Staphylococcus aureus* ATCC 25923	
	R	R (%)
PLA	/	/
PLA + 2%CH	>4.80	100.00
PLA + 2%QCH	>4.80	100.00
PLA + 2%CH + CHP’sBBL	3.34	99.95
PLA + 2%CH + QCHP’sBLL	2.90	99.87
PLA + 2%QCH + CHP’sBBL	1.36	95.64
PLA + 2QCH + QCHP’sBBL	3.65	99.98
PLA + 2%CH + CHP’sJUN	3.32	99.95
PLA + 2%CH + QCHP’sJUN	4.19	99.99
PLA + 2%QCH + CHP’sJUN	4.55	100.00
PLA + 2%QCH + QCHP’sJUN	4.11	99.99

R—antimicrobial activity; R (%)—antimicrobial activity in percentages.

## Data Availability

The data presented in this study are available in the article, or on demand from the corresponding authors.

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
