# Peer review of "Biomass-Derived Plant Extracts in Macromolecular Chitosan Matrices as a Green Coating for PLA Films"

_jfb, 2022, doi:10.3390/jfb13040228_

Round 1
Reviewer 1 Report
Manuscript contains enough scientific finding with proofs, recommend for publication.
Author Response
Thank you very much for such a nice comment.
Reviewer 2 Report
The authors present a method to form chitosan-plant extracts composite apply it as an active material for PLA film coating. Consequently, the functionalized PLA film has antifogging, antioxidant and antimicrobial properties. Overall, the work shows an interesting idea to fabricate environmental friendly functional composite based on biomass-derived materials. However, there are some concerns before accepting the manuscript.
1. The abstract section includes too many detailed experimental design and results, thus becoming quite lengthy. The authors should shrink the abstract to make it focus on 1) existing technological gap in the field, 2) novelty of your proposed solution, and 3) clear outcomes.
2. For the introduction section, it focuses too much on the "big picture" of the biomass and package field with very few scientific comments on the recent technical progress on these fields. The authors need to include more state-of-the-art works and provide comments on them.
3. How efficient the plant extracts-based composite synthesis is by using chitosan particles compared with other existing methods? Can we quantify it?
4. For the plant extracts-chitosan coating of the package, the authors didn't characterize the stability of the film adherence on PLA (e.g., under sonication, fiction with other surfaces, water flushing etc.,) Please provide additional experiments to quantify it.
Reviewer 3 Report
This submission by Zemljic et al. pursuing ‘green’ 2-layer chitosan-based coatings on PLA films matching and hopefully exceeding the properties of conventional package materials is an extremely important research goal. Authors have also improved upon their previous work by implementing quaternary chitosan, for pH-resistant activities.
Due to excellent nature of this work, this reviewer recommends publication of this submission with a few minor suggestions.
1. Surface property studies of the various formulations are very well designed, and results are reasonable and discussion sound. It is interesting, although not surprising to see highest hydrophilicity being observed for quaternary chitosan coatings.
2. Please revisit all text and revise as necessary. Some typos while minor in nature are distracting. For instance, in line 124 please change ‘physic-chemical’ to physio-chemical and so on.
3. While it is clear that 2-layer coating is successful by showing improved inhibition against oxidation from more reliable ABTS method as authors have explained very well. The 2-layer samples PLA+2%QCH+CHP’sJUN and BBL showing worse performance is somewhat of a bottleneck in the application of this study, as seen from the inhomogeneity and aggregation of the deposited film. Authors may want to explore alternate methods for attaining homogeneity for such bi- or may be tri-component coating in their future studies. While this clearly doesn’t affect the impact of this work, synthetic aspects can be improved upon and clarified upon even in this work. Beyond refining ion gelation technique employed here authors should also mention manufacturer and model number of the printing table used as PLA film formulation technique may not be clear to large audience. ‘Rolling speed level 3 and magnet power level 3’ are very vague description to say the least. The homogeneity of deposited form also plays a role in giving large trial errors during their antimicrobial studies as it has been noted.
4. Authors are pointing towards minimal leaching occurring based on their FTIR studies. While it is visibly not hard to qualitatively agree that the peak intensities at ~1080 and 1750 are decreasing it will be more convincing if authors have in-tandem quantitative analysis such as quite simply just the relative intensity ratios. This is not a major issue however as the coating integrity has remained unchanged upon the desorption studies.
